# Prognostic Value and Immunohistochemical Analysis of Mismatch Repair Deficiency in Patients with Stage II and III Colorectal Carcinoma—A Single-Center Study

**DOI:** 10.3390/medicina56120676

**Published:** 2020-12-08

**Authors:** Tijana Denčić, Miljan Krstić, Aleksandar Petrović, Maja Jovičić-Milentijević, Goran Radenković, Marko Jović, Nikola Živković, Sonja Šalinger-Martinović, Branko Branković, Simona Stojanović

**Affiliations:** 1Department of Pathology, Faculty of Medicine, University of Niš, Clinical Centre Niš, 18000 Niš, Serbia; krstic.miljan@gmail.com (M.K.); majajmi@gmail.com (M.J.-M.); nikolazivkovich@gmail.com (N.Ž.); 2Department of Histology and Embryology, Faculty of Medicine, University of Niš, 18000 Niš, Serbia; aleksandar.petrovic@medfak.ni.ac.rs (A.P.); goran.radenkovic@medfak.ni.ac.rs (G.R.); markojovic10@yahoo.com (M.J.); 3Department of Internal Medicine and Patient Care, Faculty of Medicine, University of Niš, Clinical Centre Niš, 18000 Niš, Serbia; sonja.salinger@gmail.com; 4Department of Surgery, Faculty of Medicine, University of Niš, Clinical Centre Niš, 18000 Niš, Serbia; mbbrankovic@gmail.com; 5Department of Oral Surgery, Faculty of Medicine, University of Niš, 18000 Niš, Serbia; tarana.simona@gmail.com

**Keywords:** adjuvant chemotherapy, mismatch repair deficiency, colorectal cancer, immunohistochemical analysis

## Abstract

*Background and objectives:* Deficient mismatch repair (MMR) status is associated with good prognosis but poor therapeutic response to adjuvant chemotherapy in patients with colorectal cancer. However, there are some opposed arguments considering therapeutic outcomes in patients with evidenced MMR deficiency in colorectal cancer. The aim of the study was the investigation of prognostic value and immunohistochemical analysis of the MMR-deficiency tumors. *Materials and Methods:* The study enrolled 104 patients with resected stage II and III colorectal cancer samples from the period 2018–2019. *Results:* The tumors with deficient MMR status were significantly associated with age up to 50 years and right-sided localization (*p* < 0.001). During the follow-up period of 22.43 ± 6.66 months, 21 patients (20.2%) died, whereas 14 patients (13.5%) had relapses. The loss of mutL homologue 1/postmeiotic segregation increased 2 (MLH1/PMS2) expression, compared to proficient MMR tumors, was associated with shorter disease-free survival in patients with lymphovascular invasion (*p* < 0.05), perineural invasion (*p* < 0.01), stage III (*p* < 0.05) and high-grade tumor (*p* < 0.05). *Conclusions:* This retrospective pilot study of a single-center cohort of patients with stage II and III colorectal cancer highlights the clinical importance of using immunohistochemistry (IHC) analysis as a guide for diagnostic algorithm in a country with limited resources, but with a high prevalence of colorectal carcinoma in the young patients. MMR-deficiency tumors compared with proficient MMR colorectal cancer was not shown to be a significant predictor of disease-free and overall survival.

## 1. Introduction

Colorectal cancer (CRC) is a leading cause of cancer-related morbidity and mortality worldwide, especially in developing countries [1]. There are different pathways of colorectal carcinogenesis. Microsatellite instability (MSI) is a unique molecular change and a hypermutable phenotype that results from the loss of protein activities responsible for repairing errors during the DNA replication (MMR-mismatch repair), leading to the accumulation of mutations and alterations in microsatellite length [2]. Patients with hereditary non-polyposis colorectal cancer or Lynch syndrome demonstrate microsatellite instability in more than 90% of cases [3], while sporadic colorectal carcinoma is registered in 15% of cases [4].

MMR-deficiency (dMMR) tumor is associated with good prognosis but poor therapeutic response to adjuvant chemotherapy in patients with colorectal cancer stage II. Adjuvant chemotherapy may offer better survival benefit in stage III colorectal cancer [5]. On the other hand, in some published studies [6,7], dMMR was not associated with a favorable outcome and no survival differences were found between patients with dMMR colorectal cancers compared with those who had MMR-proficiency tumors (pMMR). However, there are some controversial arguments considering therapeutic outcomes in patients with evidenced dMMR tumors.

The College of American Pathologists recommended an initial evaluation for MMR deficiency by immunohistochemistry (IHC) testing with a four-antibody panel including MLH1, MSH2, MSH6, and PMS2. IHC testing for protein expression of the four MMR genes is a cheap, fast, and simple testing in everyday practice.

The aim of the study was the investigation of immunohistochemical analyses of dMMR tumors and the evaluation of the prognostic significance of MMR deficiency in paraffin-embedded tumor blocks of resected colorectal cancer stage II and III.

## 2. Materials and Methods

The study was conducted at the Clinical Center Niš, Republic of Serbia, and included 104 patients younger than 75 years of age, with resected colorectal carcinoma stage II and III. Patients with colorectal cancer who were treated with neoadjuvant chemotherapy and radiation therapy were excluded. The registry of pathological reports from the archive of the Pathology and Pathological Anatomy was used for the selection of formalin-fixed paraffin-embedded blocks with colorectal carcinoma samples collected in the period from 2018 to 2019. Follow-up data for these patients was collected until August the 10th 2020. This study obtained the approval of the local Ethical Committee No.:12-3340-2/1 (approval date: 26.05.2020). All experiments were performed in accordance with the relevant guidelines.

The histopathology reports comprised the following data: histopathological type of tumor, localization, low or high histological grade, pronounced or poor peritumoral lymphocytic infiltrate, positive or negative Crohn’s-like reaction, present or absent mucin secretion, and pathological stage.

Patients were distributed into two groups according to age—those younger than 50 and those over 50 years of age. Tumors localized in the cecum, ascending colon, and transverse colon are classified as right-sided, while tumors localized in descending colon, sigmoid colon, or the rectum are considered left-sided tumors. The tumor invasion depth, lymph nodes, and metastases status were used for pathological staging (TNM), as defined by the American Joint Committee on Cancer (AJCC). Crohn’s-like reaction, characterized by the presence of the lymphoid aggregates, of which some resemble the morphology of a lymph follicle, is present on the tumor periphery. Crohn’s-like reaction was reported as positive (the presence of at least two lymphoid follicles with germinal centers surrounding the tumor) or negative. Pronounced peritumoral lymphocytic infiltrate was a prominent lymphocytic reaction at the growing tumor margin without lymphoid aggregates. Other inflammatory cells were also present. Tumor budding was defined as single tumor cells or cluster of less than four cells at the invasive front of the tumor. The quantification of tumor budding was assessed on hematoxylin and eosin stained slides at magnification ×200. The specimen with the highest number of tumor budding was considered a hotspot (in a field measuring 0.785 mm 2). We used a two-tier system: low-grade tumor budding (0–9 tumor buds) and high-grade tumor budding (≥ 10 tumor buds). Adjuvant chemotherapy with 5-fluorouracil (5-FU) was given to patients with stage II and III tumors, following the standard schedules and doses.

All immunohistochemistry analyses were done manually and performed in the laboratories at the Department of Histology and Embryology, Faculty of Medicine, University of Niš, Serbia. The tissues slides were dewaxed by heating at 55 °C for 30 min and by three washes, 5 min each with xylene. Tissues were rehydrated by a series of five-minute washes in 100%, 95%, and 80% ethanol and distilled water. Endogenous peroxidase activity was blocked with 3% hydrogen peroxide for ten minutes. The antigen was unmasked in a water bath for 30 min and the samples were incubated with the primary antibody at room temperature for 40 min. The samples were then incubated with secondary antibody for 20 min, each at room temperature. The monoclonal antibodies (mutL homologue 1-MLH1, clone ES05; postmeiotic segregation increased 2-PMS2, clone EP51; mutS homologue 2-MSH2, clone FE11; and mutS homologue 6-MSH6, clone EP49) were manufactured by the same company (Dako, Glostrup, Denmark) in the form ready-to-use. The system DAKO En Vision with chromogen diaminobenzidine (DAB) was used for visualization. Counterstaining was performed with Mayer’s hematoxylin solution. After rewashing the slides in water for 5 minutes, we finally dehydrated, cleared, and mounted the sections. The slides were ready for microscopic observation. Microscopic images of pathohistological specimens were obtained at the Department of Histology and Embryology, Faculty of Medicine, Niš by recording tissue specimens on the optical microscope (Olympus BΧ50, Tokyo, Japan), equipped with digital camera DFC 295 (Leica Microsystems, Wetzlar, Germany). Immunohistochemical markers of the presence or absence of nuclear immunoreactivity of tumor cells were determined by using a positive internal control for each specimen: unchanged colorectal epithelium, lymphocytes and stromal cells. The presence of nuclear expression was regarded as a positive result (proficient MMR), while nuclear immunoreactivity loss of one or more MMR proteins was regarded as a negative result (deficient MMR).

## Statistical Analysis

Statistical analysis was performed using the software SPSS ver. 17.0 (SPSS Inc, Chicago, IL, USA). Categorical variables are given as frequencies (n) and percentages (%). Comparison of the distribution of clinical and histopathological features between the examined groups was performed using the Chi-Square and Fisher’s exact test. Analysis of the relationship between MMR-deficient cancers and factors of interest was performed by univariate and multivariate logistic regression analysis (the odds ratio value (OR) and its 95% confidence interval—95% CI were calculated). Statistical significance was defined as *p* < 0.05.

## 3. Results

The loss of nuclear expression of at least one MMR protein was recorded in 12 (11.54%) cases. The loss of expression of heterodimer pair MLH1/PMS2 protein was found in 7 (6.73%) patients (Figure 1), while the loss of MSH2/MSH6 and MSH6 expression was evidenced in 1 (0.96%) patient. An isolated loss of PMS2 expression was present in 3 (2.88%) cases. The isolated loss of nuclear expression of MLH1 and MSH2 protein was not detected.

Demographic, clinical, and histopathological features of patients are given in Table 1.

Univariate logistic regression analysis showed that significant predictors of dMMR were: age ≤50 years, right-sided localization as well as the positive Crohn’s-like reaction and high grade tumor budding (Table 2). Expression of the following factors points to a higher probability of dMMR presence: age ≤50 years (*p* < 0.001), right-sided tumor localization (*p* < 0.001,) Crohn’s-like reaction (*p* < 0.001), and high-grade tumor budding (*p* < 0.001). There was no statistically significant association between the dMMR and the following parameters: gender, tumor type, mucin secretion, peritumoral lymphocytic infiltration, tumor grade, lymphovascular and perineural invasion, 5-FU based chemotherapy, and tumor stage.

Using a multivariate logistic regression analysis, age ≤50 years and the right-sided localization of tumor were shown as the most important predictors of dMMR tumors (*p* < 0.001 and *p* < 0.01, respectively) (Table 2).

Follow-up data for the whole group of patients was collected until August 10, 2020. During the follow-up period of 22.43 ± 6.66 months, 21 patients (20.2%) died, and 14 patients (13.5%) had relapses. Demographic, clinical, and histopathological features according to relapse occurrence are presented in Table 3. Relapses were more frequent in patients with high-grade tumor (*p* < 0.05).

Using a univariate Cox regression analysis, a possible predictor of disease-free survival was identified-the presence of a high-grade tumor. In the multivariate model (χ^2^ = 8.426, *p* < 0.05), none of the variables was shown to be an independent predictor of disease-free survival (Table 4).

MMR-deficiency tumors compared with proficient MMR colorectal cancer was not shown to be a significant predictor of disease-free survival. In colorectal carcinoma, MMR-proficiency tumors were predictors of longer disease-free survival compared to any loss of MMR protein expression (*p* < 0.05), especially MLH1/PMS2 loss (*p* < 0.01).

The loss of MLH1/PMS2 expression, compared to pMMR, was associated with shorter disease-free survival in patients with lymphovascular invasion (*p* < 0.05), perineural invasion (*p* < 0.01), stage III (*p* < 0.05), and high-grade tumor (*p* < 0.05)—Figure 2.

Demographic, clinical and histopathological features according to life status are given in Table 5. Death event was associated with mucinous secretion (*p* < 0.05), lymphovascular invasion or stage III (*p* < 0.05), high-grade tumor (*p* < 0.001), and previous relapse (*p* < 0.001).

Even though a univariate Cox regression analysis has identified a number of possible predictors of the overall survival in the studied group, the only independent predictors in the multivariate model (χ^2^ = 27.196, *p* < 0.001) were older age (*p* < 0.05) and high-grade tumor (*p* < 0.05) (Table 6).

## 4. Discussion

The molecular phenotype of colorectal carcinoma is closely related to demographic, clinical, and pathohistological features; biological behavior; prognosis; and therapy response [2]. Previous studies [8,9,10,11,12,13] showed different features of dMMR tumors, such as female gender, younger age of patients, poor differentiation, presence of extracellular and intracellular mucin, proximal localization in the colon, Crohn’s-like reaction, and pronounced lymphocytic infiltration.

Our results show that right-sided colon cancers are statistically significantly associated with dMMR, which is in keeping with literature data [4,8,9,10,11,12,13]. The association between younger age of a patient and the loss of MMR proteins expression by IHC was also markedly evident and consistent with literature data [4,8,9,10,11,12,13]. A statistically significant association between MMR deficiency and younger age of patients and right-sided tumor localization is another in a series of diagnostic parameters for the approach to a specific group of patients with CRC.

The use of IHC testing for deficient MMR to document protein expression in our center is not available despite international guidelines. The prevalence of dMMR in our study is relatively lower in comparison to literature data, but previous studies in Japan [13] and the USA [14] reported similar results a deficient MMR tumor prevalence of 8.4% and 10.7%, respectively. The most frequent expression pattern in our study was a combined loss of MLH1 and PMS2 proteins, with 6.73% of the overall tissue sample, which is also consistent with previous literature data [11]. In the Pakistan population, colorectal cancer with dMMR comprised 34% of the overall tissue sample [8], 33% in Egypt [11], 21% in Singapore [15], and 15% in western studies [16].

The most commonly used adjuvant chemotherapy for the treatment of CRC is the 5-FU. Since the initial discovery of MSI in CRC, a number of studies have confirmed a better outcome of patients with MMR-deficiency CRC in stage II and III compared with patients who have MMR-proficiency tumors [17,18]. Sinicrope et al. observed that, compared with patients who have proficient MMR tumors, patients with MMR-deficiency tumors experienced statistically significant improvements in disease-free and overall survival [19]. An association of dMMR tumors with improved outcomes was observed in patients with stage II and III but was statistically significant only in stage III when dMMR was compared with pMMR tumors [20,21]. A meta-analysis of patients with stage II and III CRC treated with 5-FU-based adjuvant chemotherapy demonstrated no survival benefit for patients with MMR-deficiency tumor [22]. Kontourakis et al. recommended adjuvant chemotherapy for all patients with high-risk stage II and stage III colorectal cancer [5]. Lamberti C et al. did not find an association of dMMR tumors with a favorable outcome compared with those who had MMR-proficiency tumors [6]. A relatively small subset of dMMR tumor cases may explain described data discrepancies and controversial arguments considering therapeutic outcomes in patients with evidenced MMR-deficiency tumors.

In our study, a median follow-up period was approximately two years (22.43 ± 6.66 months), and during this time 21 patients (20.2%) died (13.5%). In our patients, population relapses were more frequent in patients with high-grade tumor (*p* < 0.05). Using a univariate Cox regression analysis, we identified a possible predictor of disease-free survival, but in the multivariate model, none of the variables was shown to be a significant independent predictor. MMR-deficiency was not approved as a significant predictor of disease-free survival. MMR-proficiency colorectal adenocarcinoma was a predictor of longer disease-free survival compared to any loss of MMR expression, especially loss of MLH1/PMS2 protein expression. The loss of MLH1/PMS2 protein expression was associated with shorter disease-free survival in patients with lymphovascular invasion, perineural invasion, stage III, and high-grade tumors, compared to MMR-proficiency tumors. Lethal outcome was associated with mucinous adenocarcinoma, perineural and lymphovascular invasion, stage III, high-grade tumor, and previous relapse. Stage III, high-grade tumor, and lymphovascular and perineural invasion are independently associated with dMMR CRC recurrence, suggesting that screening for this factor in routine clinical practice may assist with adjuvant chemotherapy decision-making. There were no statistically significant differences in the overall survival between dMMR and pMMR tumors. The only independent predictors in the multivariate model were greater age and high tumor grade in survival times.

This retrospective study had a few limitations: small number of patients and unavailable IHC screening for MMR-deficiency in everyday, routine practice. There were no available data on comorbidities that may influence the outcome.

## 5. Conclusions

This retrospective pilot study of a single-center cohort of patients with stage II and III colorectal cancer highlights the clinical importance of using IHC analysis as a guide for diagnostic algorithm in an early stage of CRC. A statistically significant association of MMR deficiency with younger patients and right-sided tumor localization is another in a series of diagnostic parameters for the approach to a specific group of patients with colorectal cancer. In a country with limited resources, but with a high prevalence of colorectal carcinoma in the young patients, IHC analysis for deficient MMR proteins is recommended. MMR-deficiency was not shown as a significant predictor of disease-free survival and overall survival in stage II and III CRC. The objective of future research is to detect biomarkers that could provide the identification of the prognostic panel of biomarkers and the characterization of predictive biomarkers to help in the selection of the most appropriate therapy.

## Figures and Tables

**Figure 1 medicina-56-00676-f001:**
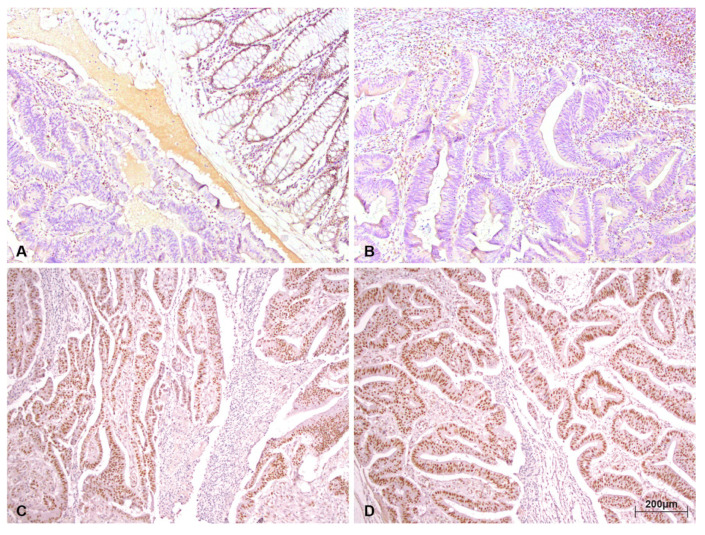
Immunohistochemical staining pattern of a deficient mismatch repair (dMMR colorectal carcinoma ×100, with the loss of MLH1 (**A**), loss of PMS2 (**B**), and intact staining of MSH2 (**C**) and MSH6 (**D**).

**Figure 2 medicina-56-00676-f002:**
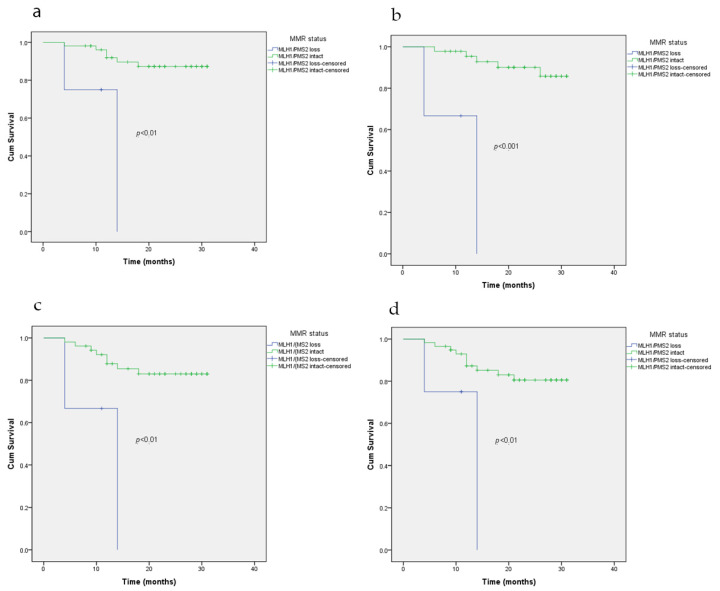
Kaplan–Meier curve of disease-free survival according to MMR status (deficient or proficient MMR) in patients with lymphovascular invasion (**a**), perineural invasion (**b**), stage III (**c**), high-grade tumor (**d**).

**Table 1 medicina-56-00676-t001:** Expression of MMR proteins in colorectal carcinoma and its association with demographic, clinical, and histopathological features.

Features	dMMR (*n* = 12)		Expression of All MMR Proteins (*n* = 92)		Total (*n* = 104)		*p* Value
Age (years)							
≤50	9	(75.00%)	6	(6.52%)	15	(14.42%)	0.0000 ***
>50	3	(25.00%)	86	(93.48%)	89	(85.58%)	
Gender							
Female	7	(58.33%)	39	(42.39%)	46	(44.23%)	0.2980
Male	5	(41.67%)	53	(57.61%)	58	(55.77%)	
Tumor localization							
Right	10	(83.33%)	22	(23.91%)	32	(30.77%)	0.0001 ***
Left	2	(16.67%)	70	(76.09%)	72	(69.23%)	
Chemotherapy given (yes)	11	(91.67%)	62	(67.39%)	73	(70.19%)	0.1031
TNM staging							
Stage III	7	58.33%	48	52.17%	55	52.88%	0.765
Stage II	5	41.67%	44	47.83%	49	47.12%	
Tumor histology							
Conventional adenocarcinoma	10	(83.33%)	66	(71.74%)	76	(73.08%)	0.5066
Mucinous adenocarcinoma	2	(16.67%)	26	(28.26%)	28	(26.92%)	
Tumor grade							
High-grade	9	(75.00%)	53	(57.61%)	62	(59.62%)	0.3525
Low-grade	3	(25.00%)	39	(42.39%)	42	(40.38%)	
Mucinous secretion							
Present	7	(58.33%)	28	(30.43%)	35	(33.65%)	0.0999
Absent	5	(41.67%)	64	(69.57%)	69	(66.35%)	
Peritumoral lymphocytic infiltrate							
Pronounced	8	(66.67%)	39	(42.39%)	47	(45.19%)	0.1137
Poor	4	(33.33%)	53	(57.61%)	57	(54.81%)	
Crohn’s-like reaction							
Positive	8	(66.67%)	13	(14.13%)	21	(20.19%)	0.0002 ***
Negative	4	(33.33%)	79	(85.87%)	83	(79.81%)	
Tumor budding							
High-grade	9	(75.00%)	17	(18.48%)	26	(25.00%)	0.0000
Low-grade	3	(25.00%)	75	81.25%	78	75.00%	
Lymphovascular invasion							
Present	8	66.67%	49	53.26%	57	54.81%	0.540
Absent	4	33.33%	43	46.74%	47	45.19%	
Perineural invasion							
Present	6	50.00%	43	46.74%	49	47.12%	10.000
Absent	6	50.00%	49	53.26%	55	52.88%	

Data were presented as frequencies (*n*) and in percentage (%)); ***—*p* < 0.001; tumor nodes metastases, TNM; deficient mismatch repair, dMMR; mismatch repair, MMR.

**Table 2 medicina-56-00676-t002:** Odds Ratio values for dMMR status prediction factors (univariate and multivariate logistic regression analysis).

**Univariate Logistic Regression**	***p* Value**	**OR**	**(95% CI of OR)**
Factor			
Age ≤50 years	<0.001	43.000	(9.157–201.922)
Female gender	0.3014	1.903	(0.562–6.443)
Right-sided tumor localization	<0.001	15.909	(3.238–78.168)
TNM stage (III)	0.688	1.283	(0.379–4.340)
Conventional adenocarcinoma	0.4018	1.970	(0.404–9.606)
High-grade tumor	0.2574	2.208	(0.561–8.692)
Present mucin secretion	0.0639	3.200	(0.935–10.954)
Pronounced peritumoral lymphocytic infiltration	0.1226	2.718	(0.764–9.673)
Crohn’s-like reaction	<0.001	12.154	(3.195–46.227)
Tumor budding (high-grade)	<0.001	13.235	(3.235–54.142)
Lymphovascular invasion (yes)	0.385	1.755	(0.494–6.238)
Perineural invasion (yes)	0.832	1.140	(0.342–3.796)
**Multivariate logistic regression (Step 2)**	***p* Value**	**OR**	**(95% CI of OR)**
Factor			
Age ≤50 years	<0.001	77.924	(8.029–756.225)
Right-sided tumor localization	0.0048	30.491	(2.843–327.060)
Crohn’s-like reaction	0.236	3.483	(0.442–27.422)
Tumor budding (high-grade)	0.280	2.872	(0.424–19.436)
Constant	<0.001	0.004	(3.195–46.227)

OR—Odds ratio, CI—Confidence interval.

**Table 3 medicina-56-00676-t003:** Demographic, clinical, and histopathological features according to relapse occurrence.

Features	Relapse	No Relapse	*p* Value
Age (years)	70.0 (62.8–73.2)	66.0 (54.0–71.2)	1.364 (0.172)
Gender (male)	6 (42.9%)	52 (57.8%)	0.572 (0.388)
Localization (right-sided)	7 (50.0%)	25 (27.8%)	1.862 (0.121)
Chemotherapy given (yes)	11 (78.6%)	62 (68.9%)	0.179 (0.547)
TNM stage (III)	10 (71.4%)	45 (50.0%)	1.456 (0.160)
Tumor histology (conventional adenocarcinoma)	8 (57.1%)	68 (75.6%)	1.257 (0.195)
Tumor grade (high)	12 (85.7%)	50 (55.6%)	3.410 (0.041)
Mucinous secretion (yes)	7 (50.0%)	28 (31.1%)	1.182 (0.224)
Crohn-like reaction (positive)	4 (28.6%)	17 (18.9%)	0.232 (0.474)
High-grade tumor budding (yes)	5 (35.7%)	21 (23.3%)	0.440 (0.332)
Lymphovascular invasion (yes)	8 (57.1%)	49 (54.4%)	0.000 (1.000)
Perineural invasion (yes)	7 (50.0%)	42 (46.7%)	0.000 (1.000)
MMR status (proficient vs deficient)	12 (85.7%)	80 (88.9%)	0.000 (0.663)
MMR status Proficient	12 (85.7%)	80 (88.9%)	2.163 (0.706)
MLH1/PMS2 loss	2 (14.3%)	5 (5.6%)	
PMS2 loss	0 (0.0%)	3 (3.3%)	
MSH2/MSH6 loss	0 (0.0%)	1 (1.1%)	
MSH6 loss	0 (0.0%)	1 (1.1%)	

**Table 4 medicina-56-00676-t004:** Cox regression analysis of disease-free survival.

	Univariate		Multivariate	
	HR (95% CI for HR)	*p* Value	HR (95% CI for HR)	*p* Value
**Tumor Grade (High)**	4.914 (1.098–21.992)	0.037	2.935 (0.556–15.499)	0.205

Hazard ratio, HR; Confidence interval, CI.

**Table 5 medicina-56-00676-t005:** Demographic, clinical, and histopathological features with reference to fatal outcome.

Features	Death	No-Death	*p* Value
Age (years)	70.0 (62.8–73.2)	66.0 (54.0–71.2)	1.364 (0.172)
Gender (male)	11 (52.4%)	47 (56.6%)	0.011 (0.808)
Localization (right-sided)	9 (42.9%)	23 (27.7%)	1.164 (0.195)
Chemotherapy given (yes)	18 (85.7%)	55 (66.3%)	2.172 (0.110)
Relapse (yes)	8 (38.1%)	6 (7.2%)	11.185 (0.001)
TNM stage (III)	16 (76.2%)	39 (47.0%)	4.624 (0.026)
Tumor histology (adenocarcinoma)	11 (52.4%)	65 (78.3%)	4.486 (0.026)
Tumor grade (high)	19 (90.5%)	43 (51.8%)	8.865 (0.001)
Mucinous secretion (yes)	10 (47.6%)	25 (30.1%)	1.581 (0.195)
Crohn-like reaction (positive)	4 (19.0%)	17 (20.5%)	0.000 (1.000)
High-grade tumor budding (yes)	3 (14.3%)	23 (27.7%)	0.975 (0.266)
Lymphovascular invasion (yes)	15 (71.4%)	42 (50.6%)	2.154 (0.140)
Perineural invasion (yes)	14 (66.7%)	35 (42.2%)	3.113 (0.053)
MMR status (proficient vs. deficient)	20 (95.2%)	72 (86.7%)	0.498 (0.452)
MMR status Proficient	20 (95.2%)	72 (86.7%)	1.553 (0.817)
MLH1/PMS2 loss	1 (4.8%)	6 (7.2%)	
PMS2 loss	0 (0.0%)	3 (3.6%)	
MSH2/MSH6 loss	0 (0.0%)	1 (1.2%)	
MSH6 loss	0 (0.0%)	1 (1.2%)	

**Table 6 medicina-56-00676-t006:** Cox regression analysis of the overall survival.

Features	Univariate		Multivariate	
	HR (95% CI for HR)	*p*	HR (95% CI for HR)	*p*
Age (years)	1.095 (1.014–1.184)	0.021	1.104 (1.013–1.204)	0.026
TNM stage (III)	3.068 (1.123–8.381)	0.029	0.789 (0.224–2.776)	0.711
Tumor grade (high)	7.575 (1.764–32.532)	0.006	7.637 (1.163–50.162)	0.034
Relapse (yes)	5.395 (2.207–13.189)	0.000	2.678 (0.983–7.294)	0.054

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
