# Peer review of "Prognostic Value and Immunohistochemical Analysis of Mismatch Repair Deficiency in Patients with Stage II and III Colorectal Carcinoma—A Single-Center Study"

_medicina, 2020, doi:10.3390/medicina56120676_

Round 1
Reviewer 1 Report
The manuscript is of interest in confirming previously reported findings with a negative result, although the authors results are from a single centre. The work is well done. However, the results provide a low priority for publication in view of a negative outcome in a single centre that merely confirms previously published findings
Author Response
We are grateful for your consideration of this manuscript, and we also very much appreciate your suggestions, which have been very helpful in improving the manuscript.
Point 1. In our pilot investigator-initiated study, we have used already presented the paraffin blocks in our data base (retrospective part). Then, we performed IHC testing, because this method is not a routine procedure in our country, due to low fund (prospective part of our study). We tried to implement guidelines suggested the diagnostic procedure for stage II, but also to look at the stage III tumor samples. We could not manage multicentric study, and we tried to emphasize the importance of this method not only in diagnostic but also in the therapeutic point of view. Adjuvant therapy was administered to almost of two third of patients, and it was useless and possibly harmful in a patients with stage II colorectal cancer. It was the reason why we decided to buy IHC markers by ourselves, and to try to make attention for this important issue.
Point 2. Comprehensive editing which includes proofreading, checking grammar, punctuation and spelling have also been done by an English-language copy editor.

Reviewer 2 Report
The authors present an unicenter retrospective work about MMR by IHQ in localized colorrectal cancer.
The concept of the article lacks of novelty. The authors mix 2 cohorts of patients completely diferent from a clinical and pronostical point of view. Patients with stage II and III CCR, are different in dMMR presence and clinical behaviour with adjuvant treatment.
Conclusion "was associated with shorter disease-free survival in patients with lymphovascular, 193 perineural invasion, stage III and high grade tumors" are the risk factors described in literature independiently of MMR status.
Moreover conclusions of this unicenter retrospective study contradict the data found in phase III prospective trials. (please see NCCN guidelines)
Author Response
Thank you for your comment. We are grateful for your consideration of this manuscript, and we also very much appreciate your suggestions, which have been very helpful in improving the manuscript.
Point 1. We tried to implement guidelines suggested the diagnostic procedure for stage II, but also to look at the stage III tumor samples. We could not manage multicentric study, and we tried to emphasize the importance of this method not only in diagnostic but also in the therapeutic point of view.
Point 2. Adjuvant therapy was administered to almost of two third of patients, and it was useless and possibly harmful in a patients with stage II colorectal cancer. It was the reason why we decided to buy IHC markers by ourselves, and to try to make attention for this important issue. Selection of patients when starting treatment is of paramount importance. Selected patients with stage II and the most patients with stage III CRC very often are treated with the same adjuvant therapy. We tried to make distinction between stage II and III in order to elucidate not useful and potentially harmful effect of adjuvant therapy (not exactly proven in our pilot study) in the group of patients with deficient MMR tumor.
Point 3. It is known fact, but in light of our first pilot IHC testing we tried to emphasize importance of this simple method unavailable in our routine practice in our country yet. Unfortunately, we cannot used IHC and PCR method in daily working schedule and we have to based our decisions on described findings.
Point 4. Comprehensive editing which includes proofreading, checking grammar, punctuation and spelling have also been done by an English-language copy editor.
We hope that we have successfully revised the manuscript according to the valuable comments of the reviewers. Please consider that due to the very dramatic situation with coronavirus pandemic in our country we are unable to perform further immunohistochemical staining, as we have ran out the supplies of the necessary markers.
We hope that we have now met the high criteria of your valuable journal. The publication of our manuscript in your journal would be of immense importance for our future professional career.

Reviewer 3 Report
This is an interesting study that investigated the immunohistochemical analysis of the dMMR and the evaluation of the prognostic significance of MMR-deficiency tumors in patients with stage II and III CRC. In general, the manuscript is well written; please revise the English language and check throughout the text for spelling errors. Perhaps, in the Abstract the aim of the study should be stated more clearly. Please be consistent when using abbreviations throughout the text and explain them when used for the first time (i.e. “IHC” in the Abstract, line 28). Please remove the first statement in the Introduction “The introduction should briefly place the study in a broad context and highlight why it is important”. The Methods section is clear; however further details should be included. Tables are detailed and helpful for the reader. The references list is quite short, I would suggest to include more updated references and further discussion on the possible role of Microsatellite instability and MMR as a biomarker for CRC (i.e. PMID: 30568941). In the conclusion paragraph it would be useful to add some statements on future perspectives and possible application of the results.
Author Response
We are grateful for your consideration of this manuscript, and we also very much appreciate your suggestions, which have been very helpful in improving the manuscript. We also thank the reviewers for their careful reading of our text.
All the comments we received on this study have been taken into account in improving the quality of the article, and we present our reply to each of them separately.
Point 1. Comprehensive editing which includes proofreading, checking grammar, punctuation and spelling have also been done by an English-language copy editor.
Point 2. In the Abstract the aim of the study is correct.
Point 3. “IHC” in the Abstract, line 28 is correct.
Point 4. The first statement in the Introduction is correct.
Point 5. Updated references is included.
Point 6. The conclusion is correct.
We hope that we have successfully revised the manuscript according to the valuable comments of the reviewers. Please consider that due to the very dramatic situation with coronavirus pandemic in our country we are unable to perform further immunohistochemical staining, as we have ran out the supplies of the necessary markers.

Reviewer 4 Report
My suggestions to Authors, 1) Title seems not satisfactory. IHC is just an experiment, not the whole Manuscript 2) Figures 2,3,4, and 5 should be in one figure after that subheading is acceptable. 3) There should be one demographic table, including all subjects. 4) Tables 2 and 4 should be more focused on findings rather than demographic details.
Author Response
We are grateful for your consideration of this manuscript, and we also very much appreciate your suggestions, which have been very helpful in improving the manuscript. We also thank the reviewers for their careful reading of our text.
All the comments we received on this study have been taken into account in improving the quality of the article, and we present our reply to each of them separately.
Point 1. Title is correct.
Point 2. Figures 2,3,4, and 5 are correct.
Point 3. We do apologize for inconveniences do to our mistake in table 1 uploading. We tried to summaries our data as you proposed. This very important investigator initiated study for our center, in terms of low fund, and voluntaristic and enthusiastic work of all participants in order to deserve attention of importance of IHC analysis. Tables 2 and 4 are correct.
Point 4. Comprehensive editing which includes proofreading, checking grammar, punctuation and spelling have also been done by an English-language copy editor.
We hope that we have successfully revised the manuscript according to the valuable comments of the reviewers. Please consider that due to the very dramatic situation with coronavirus pandemic in our country we are unable to perform further immunohistochemical staining, as we have ran out the supplies of the necessary markers.
We hope that we have now met the high criteria of your valuable journal. The publication of our manuscript in your journal would be of immense importance for our future professional career.

Round 2
Reviewer 2 Report
I have no new comments to authors